

# Attitude towards healthy nutrition and mental toughness: a study of taekwondo athletes

Arif Özsarı[1], Mehmet Kara[1], Ahmet Naci Dilek[2], Halil Uysal[3], Tolga Tek[4] and Şekip Can Deli[5]

[1] Faculty of Sport Sciences, Mersin University, Mersin, Turkey
[2] Faculty of Sport Sciences, Bartin University, Bartin, Turkey
[3] Institute of Education Sciences, Mersin University, Mersin, Turkey
[4] Faculty of Sport Sciences, Selcuk University, Konya, Turkey
[5] Institute of Social Sciences, Muş Alparslan University, Muş, Turkey

Corresponding author
Arif Özsarı, arifozsari@mersin.edu.tr

## ABSTRACT

Healthy nutrition is widely considered the cornerstone of optimal athletic performance, both physically and mentally. This study investigates the critical role of healthy nutrition in shaping the physical and mental performance of athletes, with a specific focus on taekwondo. This research aims to explore the potential relationship between taekwondo athletes' attitudes towards healthy nutrition and their mental toughness. The research group comprised 276 active and licensed taekwondo athletes who voluntarily participated in the study (Age M = 17.18 ± SD = 7.13, N = 125 women, N = 151 men). Ethical approval for the research was obtained prior to the commencement of the study. Data collection instruments included the Healthy Nutrition Attitude Scale, Mental Toughness Scale, and a personal information form. Confirmatory factor analysis was conducted to confirm the validity and reliability of the scales. Descriptive statistics, correlation analysis, and regression analysis were performed to explore the relationship between the variables within the research model. The correlation analysis identified two significant, positive, and moderate correlations: (1) between knowledge about nutrition and mental toughness (*r = 0.626*) and (2) between positive nutrition and mental toughness (*r = 0.672*). The regression analysis revealed that both knowledge about nutrition (*β = 0.360*) and positive nutrition (*β = 0.461*) significantly contribute to mental toughness. The findings suggest that as athletes' knowledge of nutrition expands and their attitudes towards healthy eating become more positive, their mental toughness also appears to improve. These results are both important and original, adding significant new insights to the existing research landscape.

# INTRODUCTION

Martial arts have captivated a global audience, reaching far beyond their combat origins (*Gubbels et al., 2016*). This popularity stems from their evolution into comprehensive mind-body practices, emphasizing personal development alongside physical skills (*Lafuente, Zubiaur & Gutiérrez-García, 2021*). Taekwondo is not solely rooted in tradition.

It has ascended to the world stage, securing its place as an Olympic sport (*Perfeito, Junior & Lourenço, 2021*), transforming into a contemporary sport with enduring appeal (*Tel, 2008*). With its deep cultural roots and emphasis on physical training, taekwondo offers a captivating blend of tradition and personal empowerment. Its accessibility lies in its simple and easily learnable movements, making it welcomed by diverse audiences (*Chun & Xu, 2022*). Beyond physical prowess, taekwondo cultivates the "fighting spirit", not for aggression, but for resilience and inner strength. Its core values of courtesy, mutual respect, and self-control instill strict self-discipline alongside personal growth. Perhaps the most profound aspect of taekwondo is its focus on conflict resolution-the ultimate goal is not to fight, but to use acquired skills to de-escalate and avoid confrontations. This philosophy, rooted in humanitarian ideals, makes taekwondo a powerful tool for promoting peace (*Dilek, 2021*). Within the realm of martial arts, it stands out for its emphasis on maintaining mental toughness at the highest level (*Aydin, 2020*).

Across the world of sports, from athletes to coaches to researchers, mental toughness is hailed as a key ingredient for success (*Crust, 2007*). It serves as a crucial element contributing to athletic achievement (*Farnsworth, Marshal & Myers, 2021*). This mental attribute empowers athletes to effectively navigate challenges during performance, manage motivation, concentration, self-confidence, emotions, and thoughts, fostering a positive mindset (*Erdogan, 2016*). The concept of mental toughness encompasses both the ability to perform under pressure and the capacity to adapt to the specific demands of a particular sport or situation (*Madrigal, Hamill & Gill, 2013*). In this context, mental toughness acts as a critical driver of performance, facilitating progress towards goals and development even in the face of stressful conditions; its influence is dynamic, exhibiting variability across both situations and timeframes (*Gucciardi et al., 2014*). Indeed, both mental activity and the cultivation of mental toughness are deemed critical determinants of peak performance and achievement (*Connaughton et al., 2008*). Athletes and individuals invested in their development actively pursue strategies to cultivate and maintain a high level of mental resilience (*Gucciardi, 2020*). Moreover, its reach extends beyond the sporting domain, permeating athletes' routine activities, from prioritizing social connections to optimizing training regimen and competition preparation, highlighting its holistic influence on an individual's well-being and personal growth (*Turkeri, Buyuktas & Ozturk, 2020*).

Building healthy eating habits is essential for all people (*Hoque, Hoque & A/P Thanabalan, 2018*). Nutrition is an important aspect of a healthy lifestyle for all individuals, including adolescents (*Ali et al., 2024*). The importance of regular physical activity and reduced sedentary behavior for promoting health and well-being is well documented (*Sandri, Cantín Larumbe & Cerdá Olmedo, 2023*). The cornerstone of a healthy lifestyle rests upon two pillars: balanced nutrition and consistent physical activity (*Akyol, Bilgic & Ersoy, 2008*). Athletes in particular require more nutrients compared to the general population (*Miyamoto & Shibuya, 2023*). For athletes, nutrition takes center stage as an integral component of their overall well-being and athletic success (*Smith & Jeukendrup, 2013*). A well-balanced diet serves as the key factor in optimizing athletic performance (*Purcell, 2013*). To achieve peak athletic outcomes, careful attention should be paid to sufficient calorie and fluid intake, alongside optimal meal timing strategies. This

emphasis on nutritional optimization transcends age limitations, with proper dietary practices proving crucial not only for athletic success but also, and perhaps more importantly, for young athletes, for fostering optimal growth, development, and overall health (*Cotugna, Vickery & McBee, 2005*).

Enhancing individuals' nutritional knowledge and fostering positive attitudes towards healthy dietary practices have been identified as key drivers of overall well-being (*Watson et al., 2009*). Nutritional literacy stands as a critical pillar in establishing and maintaining healthy eating habits, particularly for athletes (*Vázquez-Espino, Rodas-Font & Farran-Codina, 2022*). Optimal athletic performance and general health, often prioritized by this population, are demonstrably influenced by adherence to specific nutritional principles, guided by comprehensive knowledge and consistent practices (*Alahmadi & Albassam, 2023*). On the other hand, malnutrition or nutritional deficiencies may contribute to negative emotional states. Therefore, fostering nutritional knowledge and implementing improved dietary practices not only enhances physical health but also safeguards mental well-being (*Firth et al., 2020*).

The increasing recognition of nutrition's multifaceted influence on both health and athletic performance has fueled a surge in research within the domain of sports nutrition (*Valenzuela, 2023*). This growing awareness is evident in the plethora of recent research into various aspects of nutritional practices and their impact on athletic outcomes in the relevant literature (*Nobari, Cholewa & Suzuki, 2023*; *Ferrada-Contreras et al., 2023*; *Serhan, Yakan & Serhan, 2023*; *Hernández-Lougedo et al., 2023*; *Cannataro, Straface & Cione, 2022*; *Tirla et al., 2022*; *Laudisio et al., 2022*; *Miyamoto, Hanatani & Shibuya, 2021*; *Reuter & Forster, 2021*; *Bentley et al., 2021*; *Miyamoto, Hanatani & Shibuya, 2020*; *Massarani et al., 2019*; *Astley, Souza & Polito, 2017*; *Close et al., 2016*; *Knapik et al., 2016*; *Tawfik, El Koofy & Moawad, 2016*; *Reed et al., 2014*; *Halson, 2014*; *Logi Kristjánsson, Dóra Sigfúsdóttir & Allegrante, 2010*). However, a critical review of this extensive body of literature reveals that there is limited research specifically examining the interplay between attitudes towards healthy nutrition and mental toughness. Therefore, the present study holds significant value for its potential to fill this gap and contribute meaningfully to the existing knowledge base through its exploration of this understudied yet crucial nexus.

## MATERIALS AND METHODS

### Ethical approval
The study was conducted in accordance with the Declaration of Helsinki Ethical approval for this research was obtained from the Osmaniye Korkut Ata University Science Scientific Research and Publication Ethics Committee (Decision Number: 2022/1/03, Date: 25/01/2022).

### Research model
This study employed a relational screening model, a research methodology designed to evaluate the presence and magnitude of change between two or more variables (*Karasar, 2019*). The specific focus of this investigation was on the potential relationship between

attitude towards healthy nutrition and mental toughness in athletes. The hypotheses created within the scope of the research model are presented below.

H$_1$: Knowledge about nutrition has positive effects on mental toughness.

H$_2$: Emotion towards nutrition has positive effects on mental toughness.

H$_3$: Positive nutrition has positive effects on mental toughness.

H$_4$: Malnutrition has negative effects on mental toughness.

## Variables

Dependent variable: Mental toughness. Independent variable: Attitude towards healthy eating.

## Research group

The primary data for this study was the number of active, licensed taekwondo athletes in Osmaniye province, which was obtained from the Provincial Directorate of Youth and Sports. At the time of the research, there were 789 active taekwondo athletes. For this study, a convenience sampling method was used to select participants from this population. The study sample comprised 276 active licensed taekwondo athletes who voluntarily agreed to participate (Age M = 17.18 ± SD = 7.13, N = 125 women, N = 151 men). Written informed consent forms were obtained from all participants who met the eligibility criteria. Individuals who declined to participate were not included in the study. Athletes aged under 10 years old were excluded from the study, as their developmental stage might limit their ability to fully understand and respond to the research questions. During a 3-month period (February 10 to May 25, 2022), researchers gathered data from active licensed taekwondo athletes in Osmaniye, Turkey.

## Eligibility criteria

1) Active taekwondo athletes: Participants must currently be engaged in taekwondo training and/or competition.

2) Valid taekwondo license: Participants must possess a current and active license issued by a recognized taekwondo governing body.

3) Age requirement: Participants must be a minimum of 10 years old.

4) Voluntary participation: Participants must freely consent to participate in the research, with full understanding of the study's purpose and procedures.

## Data collection tools

*Healthy Nutrition Attitude Scale*: The scale is a 5-point Likert-type instrument developed by *Tekkursun-Demir & Cicioglu (2019)*. Comprising 21 items, the scale is structured into four distinct dimensions: knowledge about nutrition, emotion towards nutrition, positive nutrition, and malnutrition.

*Mental Toughness Scale:* To assess the participants' mental toughness levels, this study utilized the Mental Toughness Scale (MTS) developed by *Madrigal, Hamill & Gill (2013)*. This 5-point Likert-type instrument features a single dimension and 11 items specifically

designed to measure mental toughness within the athletic context. The scale employed in this study was its Turkish language version adapted by *Erdogan (2016)*.

## Data analysis

SPSS and AMOS (IBM Corp., Armonk, NY, USA) software packages were used to analyze the data. The study employed various statistical techniques to assess the validity and reliability of the scales used and to investigate the hypothesized relationships between them:

- Missing values: Examination of the data revealed no missing values, ensuring dataset completeness.
- Normality assumption: Skewness and kurtosis values were analyzed to verify the assumption of normal distribution. Results confirmed that the data follows a normal distribution.
- Multicollinearity: Tests were conducted to check for multicollinearity among variables. No significant multicollinearity issues were found.
- Confirmatory factor analysis (CFA): CFA was conducted to examine the factor structure of each scale, evaluating whether the measured items adequately reflected the underlying theoretical constructs.
- Internal consistency: Cronbach's alpha coefficients were calculated to assess the internal consistency of each scale.
- Descriptive statistics: Descriptive statistics, such as means, standard deviations, and frequency distributions, were computed for all variables to provide a basic understanding of the data.
- Pearson correlation analysis: This bivariate analysis was used to examine the linear relationships between the two constructs of interest, namely attitude towards healthy nutrition and mental toughness
- Multiple regression analysis: This multivariate technique was employed to further investigate the influence of attitude towards healthy nutrition on mental toughness, while controlling for any other relevant variables.
- The meticulous research design resulted in a pristine dataset free from missing data, bolstering the confidence in the findings.
- To prioritize data quality and accuracy, the study focused on objective measures and standardized procedures rather than relying solely on subjective responses.

## RESULTS

This section of the study presents the results of the confirmatory factor analysis, as well as the outcomes of correlation and regression analyses for the scales employed.

In Table 1 CFA is used to verify the fit of the predicted structure of the scale with the data (*Gurbuz & Sahin, 2017*). As a result of the confirmatory factor analysis of the used in the study, the model fit criteria were reviewed. According to the results of the analysis, the following to values were reached: Mental Toughness Scale: CFA for the mental toughness

**Table 1 Confirmatory factor analysis for the data collection tools.**

|  | CMIN/DF ($x^2$/df) | GFI | AGFI | CFI | IFI | RMSEA |
|---|---|---|---|---|---|---|
| Mental toughness scale | 1.370 | 0.940 | 0.906 | 0.968 | 0.969 | 0.047 |
| Healthy nutrition attitude scale | 2.017 | 0.900 | 0.860 | 0.900 | 0.902 | 0.061 |

**Table 2 Results of correlation analysis.**

| N = 276 | M | SD | 1 | 2 | 3 | 4 | 5 |
|---|---|---|---|---|---|---|---|
| 1. Knowledge about nutrition | 4.12 | 0.816 | – | | | | |
| 2. Emotion towards nutrition | 3.04 | 0.892 | −0.046 | – | | | |
| 3. Positive nutrition | 3.83 | 0.885 | 0.579** | −0.072 | – | | |
| 4. Malnutrition | 2.73 | 1.067 | −0.157** | 0.606** | −0.080 | – | |
| 5. Mental toughness | 3.98 | 0.740 | 0.626** | −0.096 | 0.672** | −0.111 | – |

Note:
** $p < 0.01$.

scale assessed the conformity criteria, yielding the following results: CMIN/DF$(x)2/df = 1.370$, CFI = 0.968, GFI = 0.940, IFI = 0.969, AGFI = 0.906, and RMSEA = 0.047. The Cronbach's Alpha ($\alpha$) value for the scale was calculated as 0.86. Healthy Nutrition Attitude Scale: CMIN/DF$(x)^2/df = 2.017$, CFI = 0.900, GFI = 0.900, IFI = 0.902, AGFI = 0.860, and RMSEA = 0.061. Internal consistency was further evaluated using Cronbach's Alpha ($\alpha$) yielding values of 0.78 for knowledge about nutrition, 0.70 for emotion towards nutrition, 0.76 for positive nutrition, and 0.80 for malnutrition. These values indicate a good degree of reliability for each dimension, suggesting they effectively measure their intended constructs. Overall, based on the CFA and Cronbach's alpha results, both the Healthy Nutrition Attitude Scale and the Mental Toughness Scale demonstrated acceptable psychometric properties for the study (*Cole, 1987*; *Segars & Grover, 1993*; *Doll, Weidong & Torkzadeh, 1994*; *Hair et al., 2014*; *Kline, 2011*).

Table 2 Pearson correlation analysis was conducted to explore the direction and strength of the relationships between the four subdimensions of the Healthy Nutrition Attitude Scale and mental toughness. The results revealed a significant positive and moderate correlation (*r = 0.626*) between knowledge about nutrition and mental toughness, suggesting that higher levels of nutritional knowledge are associated with greater mental resilience among athletes. An even stronger significant, positive, and moderate correlation (*r = 0.672*) was found between positive nutrition and mental toughness, indicating that a positive attitude towards healthy eating practices is more strongly associated with mental toughness than simply factual knowledge about nutrition. These findings suggest that individuals with stronger knowledge about nutrition tend to exhibit more positive attitudes towards healthy eating, which may further contribute to enhanced mental toughness.

The score of one variable is used to predict the other in regression analysis (*Tabachnick & Fidell, 2015*). Table 3 presents the results of the multiple regression analysis conducted

**Table 3 Multiple linear regression analysis.**

| Model | B | Std. Error | Beta (β) | t | p | VIF |
|---|---|---|---|---|---|---|
| (Constant) | 1.272 | 0.206 | – | 6.177 | 0.000 | – |
| Knowledge about nutrition | 0.326 | 0.047 | 0.360 | 7.006 | 0.000 | 1.546 |
| Emotion towards nutrition | −0.046 | 0.043 | −0.056 | −1.069 | 0.286 | 1.598 |
| Positive nutrition | 0.385 | 0.042 | 0.461 | 9.075 | 0.000 | 1.513 |
| Malnutrition | 0.011 | 0.037 | 0.016 | 0.302 | 0.763 | 1.630 |
| R = 0.734 | $R^2$ = 0.538 | Adj. $R^2$ = 0.531 | | | | |
| $F_{(4-271)}$ = 78.971 | p = 0.000 | D-W = 2.027 | | | | |

**Note:**
Dependent variable: mental toughness (MT).

to investigate the influence of the four subdimensions of the Healthy Nutrition Attitude Scale (independent variables) on mental toughness in sports (dependent variable). The Durbin-Watson values (D-W = 2.027) between 1.5 and 2.5 displayed in the table confirm the absence of multicollinearity concerns among the independent variables (*Zakerian & Subramaniam, 2009*; *Kalaycı, 2018*). Furthermore, the variance inflation factor (VIF) values fall within the acceptable range reported in relevant literature, further supporting the absence of multicollinearity issues (*Mertler & Vannatta Reinhart, 2017*). Analysis of the multiple regression model revealed its suitability for reliable interpretation of the results. The model demonstrated statistical significance ($F_{(4-271)}$ = 78.971). The model's adjusted $R^2$ value of 0.531 indicates that 53.1% of the variance in the dependent variable, mental toughness in sports, is explained by the independent variables related to attitudes towards healthy nutrition. Further investigation into the specific effects of the subdimensions revealed that both knowledge about nutrition (β = 0.360) and positive nutrition (β = 0.461) have significant positive contributions to Mental Toughness. These findings suggest that a one-unit increase in knowledge about nutrition is associated with a 0.360 increase in mental toughness, while a one-unit increase in positive nutrition is associated with a 0.461 increase in mental toughness. Therefore, these results provide support for hypotheses $H_1$ and $H_3$, which posited positive relationships between knowledge about nutrition, positive nutrition, and mental toughness in sports.

## DISCUSSION

This study offers a unique contribution by examining the specific relationship between nutritional knowledge, positive attitudes towards healthy eating, and mental toughness in taekwondo athletes. The findings suggest that promoting nutritional awareness and positive dietary habits may be a novel avenue for enhancing athlete well-being. The results of the correlation analysis revealed a significant positive and moderate correlation between both knowledge about nutrition and positive nutrition with mental toughness (Table 2). This suggests that higher levels of both nutritional knowledge and positive attitudes towards healthy eating are associated with enhanced mental resilience in athletes. Additionally, a positive correlation was observed between knowledge about nutrition and positive nutrition themselves, indicating a potential interrelationship between these

constructs. Multiple regression analysis, with the four subdimensions of the Healthy Nutrition Attitude Scale serving as independent variables and mental toughness as the dependent variable, yielded a statistically significant model (Table 3). Both knowledge about nutrition and positive nutrition emerged as significant positive predictors of mental toughness, suggesting that a one-unit increase in either variable is associated with a corresponding increase in mental resilience. The findings provide robust support for the research model, formulated under hypotheses $H_1$ (knowledge about nutrition has positive effects on mental toughness) and $H_3$ (positive nutrition has positive effects on mental toughness). In other words, research results show that improved nutritional knowledge and a positive attitude towards healthy eating directly contribute to stronger mental resilience in athletes.

The present study's results align with existing literature suggesting a positive association between nutritional knowledge, positive attitudes towards healthy eating, and mental toughness in athletes. *Ozsari, Kara & Kara (2021)* observed similar relationships in athletes across various disciplines, demonstrating that increased knowledge and positive nutrition attitudes correlate with enhanced mental toughness. *Sunuwar et al. (2022)* also found a positive correlation between nutritional knowledge and athletic performance in taekwondo athletes, further reinforcing the notion that dietary awareness and healthy eating practices contribute to both mental and physical well-being in athletes. Research findings by *Rampersaud et al. (2005)* showed that a healthy diet can improve cognitive function.

The importance of a robust understanding of nutrition for maintaining a balanced and healthy diet is well-recognized (*Calella, Iacullo & Valerio, 2017*). This applies especially to athletes, for whom knowledge about proper nutrition is considered crucial for optimizing both physical and mental performance (*Rogers & Lloyd, 1994*). Indeed, the nutritional status of individuals can significantly impact their mental performance, as it provides the essential substrate for optimal brain function and supports a variety of cognitive processes (*Bodnar & Wisner, 2005*). Further solidifying this connection, *Dunn, Turner & Denny (2007)* argue that nutritional behaviors play a key role in influencing athletes' health and athletic performance. Likewise, *Larson & Yousafzai (2015)* have demonstrated that even subtle nutritional interventions can have noticeable effects on mental development.

The role of mental toughness as a crucial multidimensional mental skill with significant implications for athletic success is well-established (*Zarei, Gharayagh Zandi & Mohebi, 2018*). Its cultivation is essential in the pursuit of optimal performance states (*Hodge, 1994*), and its link to perseverance and achievement in performance domains is undeniable (*Powell & Myers, 2017*). Extensive research supports these assertions. An empirical positive correlation between mental toughness and successful sports performance has been observed across diverse contexts. *Sheard (2009)*, found such a relationship among elite-level rugby league players, while *Joshi & Kalode (2019)* and *Guszkowska & Wojcik (2021)* reported similar findings for taekwondo athletes and athletes in various sports, respectively. Furthermore, *Bisri, Saputri & Chusniyah (2022)* identified a strong positive correlation between athletes' mental toughness and their performance, while *Beckford et al. (2016)* observed higher levels of mental toughness in elite compared to sub-elite sprinters.

Similarly, *Kuan & Roy (2007)* found a connection between mental toughness and superior performance and medal wins in wushu athletes. This consistent pattern is further reinforced by numerous studies demonstrating the superior achievement levels of mentally stronger athletes (*Weissensteiner et al., 2012*; *Hagag & Ali, 2014*; *Slimani et al., 2016*; *Zandi & Mohebi, 2016*; *Cowden, 2017*; *Giles et al., 2018*). The overwhelming evidence suggests a strong and reliable association between enhanced mental toughness and overall athletic success.

Previous work in this field highlights the crucial role of nutritional knowledge in influencing athletic success. *Davar (2012)* and *Sunuwar et al. (2022)* emphasize the importance of accurate nutritional knowledge as a fundamental determinant of athletic achievement. *Teo, Burns & Kawabata (2022)* provide empirical support for this notion, demonstrating a positive correlation between netball players' level of nutrition knowledge and their feeding behavior. However, *Bezci et al. (2018)* found that taekwondo athletes often engage in detrimental nutritional practices due to their insufficient knowledge about athlete-specific dietary needs.

Further research reinforces the link between nutritional knowledge and athletic performance. *Brum, Franchini & Moura (2023)* identified a connection between futsal players' dissatisfaction with their performance and their deviation from healthy eating habits. Similarly, *Finamore et al. (2022)* found that non-professional athletes often lack adequate nutritional knowledge. *Staśkiewicz et al. (2023)* further contribute to this discussion by revealing that athletes with a greater awareness of their nutritional needs could experience less muscle loss.

Adopting an appropriate nutritional approach is demonstrably linked to both enhanced health and improved athletic performance in athletes. Studies by *O'Brien, Collins & Amirabdollhian (2021)* and *Iwasa-Madge & Sesbreno (2022)* have revealed a direct correlation between athletes' nutritional styles and outcomes in both health and sporting settings. Recognizing the critical role of nutrition in athletic success, *Finamore et al. (2022)* emphasizes the compelling need for research aimed at bolstering the nutritional knowledge of sports participants. This research findings further solidify this connection by demonstrating the positive influence of positive nutritional attitudes on athletes' mental toughness. This suggests that enhanced knowledge about nutrition and the adoption of positive dietary practices can plausibly lead to increased mental resilience among athletes. These results highlight the interconnectedness of nutrition, mental resilience, and overall athletic well-being. Therefore, the findings have valuable implications for understanding and supporting athlete well-being.

## CONCLUSIONS

In conclusion, this research suggests that prioritizing nutrition in athlete development programs could have far-reaching benefits for their mental resilience. It opens up new avenues for exploring nutritional interventions as a means of supporting athletes' mental well-being and optimizing performance outcomes.

Future research directions could involve exploring the interplay between nutritional knowledge, attitudes, and mental toughness within diverse athletic contexts. Investigating

these relationships in both individual and team sports across various disciplines would provide a more comprehensive understanding of the potential benefits of nutritional interventions for enhancing athletes' mental resilience and overall well-being.

### Practical implications

Coaches, sports psychologists, and sports organizations play a crucial role in promoting healthy eating habits in athletes. Integrating nutritional education into training programs, offering workshops focused on sports nutrition, and promoting healthy eating from a young age within schools and sports clubs can significantly enhance athletes' mental resilience and support their overall development. All stakeholders in sport, especially policymakers, should prioritize initiatives that champion healthy eating among athletes throughout their formative years. Incorporating nutrition education into existing training frameworks is a critical step in fostering mental resilience and optimizing performance outcomes in athletes.

### Limitations

**Generalizability:** The study results are limited to taekwondo athletes and may not be generalizable to other athlete populations or the general public. Factors specific to taekwondo culture or training might influence the relationship between nutrition knowledge, attitudes, and mental toughness, making it difficult to extrapolate the findings to other contexts.

**Selection bias:** Although no intentional bias was mentioned, the convenience sampling method used to recruit participants could potentially introduce selection bias. The athletes who chose to participate might be more interested in nutrition or have different characteristics than the broader taekwondo population.

**Confounding variables:** The study design might not account for potential confounding variables that could influence the observed relationship between variables. For example, physical activity level, injury history, or socioeconomic status could influence both nutrition knowledge and mental toughness, making it difficult to isolate the specific effect of one on the other.

**Temporal limitations:** The study was conducted within a specific timeframe and may not reflect changes in taekwondo training, athlete demographics, or nutritional trends over time.

**Cultural limitations:** The research was conducted in Osmaniye, Turkey, and the cultural context might influence factors like attitudes towards healthy eating and mental toughness. Applying the findings to athletes from different cultural backgrounds might require caution.

## ACKNOWLEDGEMENTS

We would like to thank the athletes and their coaches who participated in the research.

### Funding
The authors received no funding for this work.

### Competing Interests
The authors declare that they have no competing interests.

### Author Contributions
- Arif Özsarı conceived and designed the experiments, performed the experiments, analyzed the data, prepared figures and/or tables, authored or reviewed drafts of the article, and approved the final draft.
- Mehmet Kara conceived and designed the experiments, performed the experiments, analyzed the data, prepared figures and/or tables, authored or reviewed drafts of the article, and approved the final draft.
- Ahmet Naci Dilek conceived and designed the experiments, performed the experiments, analyzed the data, prepared figures and/or tables, authored or reviewed drafts of the article, and approved the final draft.
- Halil Uysal conceived and designed the experiments, performed the experiments, analyzed the data, prepared figures and/or tables, authored or reviewed drafts of the article, and approved the final draft.
- Tolga Tek conceived and designed the experiments, performed the experiments, analyzed the data, prepared figures and/or tables, authored or reviewed drafts of the article, and approved the final draft.
- Şekip Can Deli conceived and designed the experiments, performed the experiments, analyzed the data, prepared figures and/or tables, authored or reviewed drafts of the article, and approved the final draft.

### Human Ethics
The following information was supplied relating to ethical approvals (*i.e.*, approving body and any reference numbers):

Ethical approval for this research was obtained from the Osmaniye Korkut Ata University Science Scientific Research and Publication Ethics Committee (Decision Number: 2022/1/03, Date: 25/01/2022).

### Data Availability
The raw data is available in the Supplemental File.

### Supplemental Information
Supplemental information for this article can be found online at http://dx.doi.org/10.7717/peerj.17174#supplemental-information.

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
