# Peer review of "Attitude towards healthy nutrition and mental toughness: a study of taekwondo athletes"

_PeerJ, doi:10.7717/peerj.17174_

## Round 0.1 · original submission · Minor Revisions

Basically, this manuscript is well-written and easy to understand. In addition, the background is well-written. However, as each reviewer pointed out, you need to describe your methodology in more detail, such as how you determined the sample size.

·

Basic reporting

Some major revisions are needed.

Experimental design

The experimental design is well structured. What were the eligibility criteria?

Validity of the findings

How was the sample determined?

There is no description of the power analysis and in which program was the power analysis done and on which test was it based, please provide full details.

It was not stated in which program the data analyzes were performed.

Parametric approaches were used without an explanation for the distribution of the data. How was normal distribution examined?

Were all assumptions met for multivariate regression analysis? These should be stated in the analysis section.

Additional comments

The discussion lacks a clear emphasis on the practical implications of the study's findings. How can coaches, athletes, or sports organizations use this information to enhance training or performance? Including practical applications adds value to the research.

·

Basic reporting

Thank you for your submitted manuscript entitled, “Attitude towards healthy nutrition and mental toughness: A study of taekwondo athletes’’. The area of the research is interesting; however, it needs a few amendments. Overall, the paper is well-written. The manuscript is well-structured, and it is easy to follow the sections.

The introduction provides a proper background of the topic and there is a well-presented novelty.
The key words should be different from the words in the title.
The quality of the images (tables) is good enough, but I don’t know if the reviewing version has lower resolution than the final version. If not, images should have better resolution in its final size.
It seems that the English is clear, but research articles usually do not use the word "we/our" and regularly use passive verbs (lines 22 and 196, as well as 265 and 325).

Experimental design

The experimental design meets the scope of the journal, and the hypotheses are well-defined and relevant to the community.
Methods and methodology are described detailed enough.

Validity of the findings

Most of the results are quite interesting and are well discussed.
In the Discussion it would be better to have seen more use of terms like 'originality' and 'significance'. Identify what is new in this study that may benefit readers or how it may advance existing knowledge or create new knowledge on this subject. There should be a clear conclusion on why the research findings are significant.

Reviewer 3 ·

Basic reporting

1. The manuscript is well-written and easy to understand
2. The background information is written in detail

Experimental design

1. Hypotheses are defined clearly, and research aims and data collection are clearly described
2. The model variables should be well-defined especially the data type, for example, the outcome mental toughness is a numerical variable or a categorical variable?
3. In the data analysis part, multiple statistical methods are mentioned. However, the assumptions should be checked before implementing those statistical models.

Validity of the findings

1. The tables show the statistical analysis results, and the results are discussed in a detailed way.
2. The conclusion is linked to the research aim, again, if the statistical analysis part can be interpreted better, the conclusion would be more solid.

---

## Round 0.2 · accepted · Accept

All reviewers assessed your manuscript as worthy of publication. I myself rate it as an excellent manuscript.

·

Basic reporting

The authors adequately addressed my comments.

Experimental design

The authors adequately addressed my comments.

Validity of the findings

The authors adequately addressed my comments.

Additional comments

My suggestion to authors is to always evaluate statistical assumptions based on hypothesis testing in their subsequent research.